# Kuwanon C Inhibits Tumor Cell Proliferation and Induces Apoptosis by Targeting Mitochondria and Endoplasmic Reticulum

**DOI:** 10.3390/ijms25158293

**Published:** 2024-07-30

**Authors:** Gangxiang Yuan, Peng Qian, Lin Chen, Ningjia He

**Affiliations:** State Key Laboratory of Resource Insects, Southwest University, Chongqing 400715, China; yuangx99@email.swu.edu.cn (G.Y.); qp125678@email.swu.edu.cn (P.Q.); cl125993@email.swu.edu.cn (L.C.)

**Keywords:** apoptosis, cancer, HeLa, kuwanon C, flavonoid, mitochondria

## Abstract

Kuwanon C is a unique flavonoid found in the mulberry family, characterized by two isopentenyl groups. While previous research has focused on various properties of kuwanon C, such as antioxidant, hypoglycemic, antimicrobial, food preservation, skin whitening, and nematode lifespan extension, little attention has been given to its potential role in oncological diseases. In this study, we investigate the antitumor effect of kuwanon C in cervical cancer cells and elucidate its specific mechanism of action. We assessed the antitumor effects of kuwanon C using various experimental techniques, including cell proliferation assay, wound healing assays, EdU 488 proliferation assay, mitochondrial membrane potential assay, ROS level assay, cell cycle, apoptosis analysis, and studies on kuwanon C target sites and molecular docking. The results revealed that kuwanon C significantly impacted the cell cycle progression of HeLa cells, disrupted their mitochondrial membrane potential, and induced a substantial increase in intracellular ROS levels. Moreover, kuwanon C exhibited notable anti-proliferative and pro-apoptotic effects on HeLa cells, surpassing the performance of commonly used antitumor drugs such as paclitaxel and cisplatin. Notably, kuwanon C demonstrated superior efficacy while also being more easily accessible compared to paclitaxel. Our study demonstrates that kuwanon C exerts potent antitumor effects by its interaction with the mitochondrial and endoplasmic reticulum membranes, induces a significant production of ROS, disrupts their normal structure, inhibits cell cycle progression, and stimulates apoptotic signaling pathways, ultimately resulting in the death of HeLa tumor cells. As an isopentenyl compound derived from *Morus alba*, kuwanon C holds great promise as a potential candidate for the development of effective antitumor drugs.

## 1. Introduction

Cancer is one of the greatest threats to human health, characterized by the uncontrolled division of mutated cells and the ability to infiltrate and destroy normal tissues. In advanced stages, cancer often metastasizes to various parts of the body, swiftly claiming lives [1]. According to the most recent data from the International Agency for Research on Cancer of the World Health Organization in 2020, there were 19.3 million new cancer patients and 9.9 million deaths worldwide. Furthermore, it is expected that by 2040, there will be a global increase to 28.4 million new cancer cases [2]. Cervical cancer has emerged as the fourth most common cancer among females, profoundly impacting reproductive health and endangering human reproduction [3]. Given the evident heterogeneity of cancer, treatment modalities vary considerably depending on tumor type, stage, and individual factors. Currently, five major cancer treatments have been developed, as reported by the National Cancer Institute and the American Cancer Society. These include surgical interventions, chemotherapy, radiotherapy, targeted therapies, and immunotherapies. Each treatment method possesses unique strengths and limitations, and in actual clinical practice, a combination of multiple modalities is typically used [4,5,6].

Cancer chemotherapy plays an important role in treating advanced tumors, serving as a primary systemic therapy approach for middle and advanced stages [7,8]. Natural products derived from plants, encompassing flavonoids, alkaloids, polyphenols, terpenoids, quinones, and various compounds from terrestrial and marine sources, have long been recognized as invaluable resources for developing anticancer drugs, garnering significant attention from researchers [9,10]. Certain plant-derived chemotherapeutic agents possess notable advantages such as multi-targeting capabilities, minimal side effects, and cost-effectiveness. Among them, paclitaxel, vinblastine [11,12], and camptothecin are commonly employed. Paclitaxel, derived from the yew tree (*Taxus Brevifolia*), has demonstrated widespread efficacy against diverse cancer types, including leukemia, breast cancer, melanoma, recurrent ovarian cancer, non-small cell lung cancer, prostate cancer, gastric adenocarcinoma, ovarian cancer, and esophageal cancer [13,14]. However, the effectiveness of paclitaxel, vinblastine, and camptothecin is compromised by the development of drug resistance and varying responses across different cancer types, limiting their application as plant-derived chemotherapeutic agents. Therefore, it remains crucial to continue the quest for more efficient, safer, and economical anticancer drugs that can benefit cancer patients. Leveraging the extensive history and clinical practice of traditional Chinese medicine spanning thousands of years offers promising prospects for discovering more effective and safe antitumor drugs.

Mulberry, an ancient traditional Chinese medicine, has been extensively documented in the medical book *Shennong Ben Cao Jing*. The book states that various parts of the mulberry tree, including leaves, fruits, branches, and bark, have medicinal properties, displaying anti-inflammatory [15], antioxidant, anti-aging [16], anti-diabetic [17], and antitumor efficacy [18]. Mulberry has been used for thousands of years of history in the maintenance of people’s health. With advancements in modern biotechnology, mulberry has been found to contain active substances that exhibit hypoglycemic, antioxidant, and anti-aging properties. However, research on its potential as an anticancer agent has been limited. To date, the primary natural compounds in mulberry with anticancer properties include morin [19], sanggenon C [20,21], morusin [22,23,24], norartocarpetin [25,26], 1-Deoxynojirimycin (DNJ), and resveratrol. Morin, a flavonoid extracted from mulberry trees, has also been reported to increase the sensitivity of the anticancer drug doxorubicin [27]. Another flavonoid called kuwanon C, characterized by two isopentenyl groups and also derived from *Morus alba*, has predominantly been studied for its antiviral [28], antioxidant [29], and anti-aging [16] properties, with limited exploration in anticancer research. Isopentenyl-substituted natural compounds demonstrate structural diversity and encompass a wide range of pharmacological activities, positioning them as a valuable resource for developing innovative drugs [30,31]. The incorporation of isopentenyl groups in natural product molecules not only enhances the structural diversity but also improves the affinity and bioavailability to the drug target due to increased lipophilicity, thus augmenting the drug-forming properties [32,33,34]. Whether the inclusion of two isopentenyl groups of kuwanon C enhances its potential as an antitumor agent remains unclear. 

Our study is the first to demonstrate the antitumor effect of kuwanon C at the HeLa cell level, which inhibits cell proliferation, induces cell cycle arrest, promotes apoptosis, and inhibits cell migration by targeting mitochondria and endoplasmic reticulum. These results illustrate the antitumor effect of kuwanon C and the underlying mechanism, suggesting potential clinical usage of the drug.

## 2. Results

### 2.1. Inhibition of HeLa Cell Viability by Kuwanon C

In the present study, we investigated the impact of kuwanon C, kuwanon T, albanin A, norartocarpetin, and morin on the viability of HeLa cells. These flavonoids were chosen due to their distinctive isopentenyl groups unique to mulberry (Figure 1A). Through cell viability analysis, we observed that kuwanon C, containing two isopentenyl groups, displayed significant antitumor effects on cervical cancer HeLa cells in a concentration-dependent manner (Figure 1B,C). Furthermore, the antitumor cell proliferation ability was found to increase in proportion to the concentration of kuwanon C. Comparatively, the flavonoids kuwanon C and kuwanon T, with two isopentenyl groups, displayed higher anti-proliferative activity against HeLa tumor cells compared to norartocarpetin and morin, which lack the isopentenyl group, and albanin A, which has only one isopentenyl group (Appendix A). We compared kuwanon C with commonly used clinical antitumor compounds paclitaxel and cisplatin at the cellular level. Remarkably, kuwanon C, an isopentenyl analog derived from mulberry, demonstrated stronger antitumor effects on tumor cell proliferation compared to paclitaxel and cisplatin, with an increase in the concentration (Figure 1C and Appendix A). Additionally, we observed inhibition of proliferation in breast cancer cells T47D, MDA-MB-231, glioma LN229 cells, as well as normal cells (Human Umbilical Vein Endothelial Cells, HUVEC) with increasing concentrations of kuwanon C (Appendix A).

### 2.2. Suppression of HeLa Cell Migration and Proliferation by Kuwanon C

In this study, we conducted a cell scratch assay to investigate the effect of kuwanon C on HeLa cell migration after 24 h of treatment. We observed that kuwanon C exhibited a concentration-dependent inhibition of HeLa cell migration (Figure 2A,B). Interestingly, the level of inhibition was higher in kuwanon C, which contains two isopentenyl groups, compared to albanin A with one isopentenyl group and norartocarpetin without an isopentenyl group. Notably, at 40 μM, albanin A promoted HeLa cell migration, while norartocarpetin had no inhibitory effect (Figure 2A,B). Furthermore, we performed a plate clone formation assay to assess the effect of different concentrations (0 μM (control), 1 μM, 2 μM, 5 μM, 10 μM, 20 μM) of kuwanon C, albanin A, and morin on HeLa cell clone formation. Our results revealed that at 20 μM, kuwanon C significantly inhibited the clone formation ability of HeLa cells, while albanin A and morin had no impact (Figure 2C–H). Additionally, we conducted an EdU 488 cell proliferation staining assay following a 24 h treatment with kuwanon C. The staining results demonstrated that kuwanon C inhibited the proliferation of HeLa cells in a concentration-dependent manner (Figure 2I). Moreover, when HeLa cells were simultaneously treated with 50 μM of kuwanon C, albanin A, and norartocarpetin for 24 h, only kuwanon C exhibited an inhibitory effect on HeLa cell proliferation (Figure 2J).

### 2.3. Induction of Reactive Oxygen Species Generation in HeLa Cells by Kuwanon C

We assessed the ROS levels in HeLa cells using a ROS assay kit following treatment with different concentrations (0 μM (control), 20 μM, and 40 μM) of kuwanon C, as well as 40 μM of paclitaxel, albanin A, and norartocarpetin for 12 h and 24 h, respectively. The results showed a significant increase in ROS levels in HeLa cells with escalating concentrations of kuwanon C, indicating an elevation in fluorescent green fluorescence associated with higher levels of reactive oxygen species (Figure 3A). We treated HeLa cells simultaneously with equal concentrations of kuwanon C, paclitaxel, albanin A, and norartocarpetin and utilized the ROS reactive oxygen species assay to evaluate the levels of reactive oxygen species. Our results revealed that the kuwanon C-treated group exhibited higher levels of reactive oxygen species compared to the groups treated with paclitaxel, albanin A, and norartocarpetin (Figure 3A,B). Furthermore, we conducted additional experiments by adding 5 mM of NAC, an antioxidant, to the kuwanon C group. Interestingly, we observed that NAC did not exhibit any inhibitory effect on the generation of reactive oxygen species (Figure 3C,D). The addition of 5 mM of the antioxidant NAC partially rescued the inhibition of HeLa cell viability by high concentrations of kuwanon C (Figure 3E). Furthermore, we examined the effect of shorter treatment times with kuwanon C, specifically 3 h, 6 h, and 12 h. Using flow cytometry, we measured intracellular reactive oxygen species levels and found that the kuwanon C-treated group exhibited higher levels of reactive oxygen species in a concentration-dependent and time-dependent manner (Figure 3F–I). Considering the close relationship and interaction between ROS and intracellular calcium ion levels, we treated cells with 60 μM concentrations of kuwanon C, albanin A, and norartocarpetin for 24 h to explore their impact on intracellular calcium ion levels. Our observations showed increased green fluorescence in HeLa cells from the kuwanon C-treated group, indicating higher intracellular calcium ion levels compared to the control and norartocarpetin-treated groups (Figure 3J).

### 2.4. Induction of Apoptosis in HeLa Cells by Kuwanon C

Following the treatment of HeLa cells with kuwanon C, we assessed the apoptosis rate using various methods, including cell morphology changes, apoptosis and necrosis staining assay, flow cytometry, real-time quantitative PCR (RT-qPCR) for mRNA expression of apoptosis-related genes, and Western blotting for protein expression of apoptosis-related genes. Additionally, we utilized transmission electron microscopy to observe the microstructural changes in HeLa cells between the control group and the group treated with kuwanon C. The findings demonstrated a decrease in the number of HeLa cells and an increase in HeLa cell apoptosis and necrosis with increasing concentrations of kuwanon C, While the same concentration of albanin A, norartocarpetin and paclitaxel treated groups had significantly lower apoptosis and necrosis of HeLa cells than kuwanon C treated groups (Figure 4A). Our observations revealed that the kuwanon C-treated cells exhibited damage to their normal structures, leading to cell death, whereas the control group displayed intact and undamaged cellular structures (Figure 4B). At a consistent concentration of 50 μM, we compared kuwanon C with albanin A containing one isoprenoid group and norartocarpetin without an isoprenoid group in the treatment of HeLa cells. Flow cytometry analysis revealed that the kuwanon C-treated group had a higher apoptosis rate compared to albanin A and norartocarpetin-treated groups (Figure 4C). Furthermore, analysis of apoptosis-related genes, including *GADD45A*, *CASP3*, *CASP4*, *PPP1R15A*, *PMAIP1*, *ATF3*, and *IFIT2*, depicted higher expression levels in the kuwanon C-treated group compared to both the albanin A and Norartocarpetin-treated groups, as well as the control group (Figure 4D,E). Western blotting results indicated that kuwanon C increased the expression level of GAADD34 and NOXA, proteins involved in apoptosis (Figure 4F).

### 2.5. Effect of Kuwanon C on Mitochondrial Membrane Potential

Changes in the mitochondrial membrane potential play an important role in apoptosis. Therefore, we examined the changes in mitochondrial membrane potential using JC-1 staining after treating HeLa cells with kuwanon C for 8 h and 24 h. The results revealed a decrease in mitochondrial membrane potential of HeLa cells with increasing concentrations of kuwanon C (Figure 5A). To further investigate the effects on mitochondrial membrane potential, we compared the effects of kuwanon C, paclitaxel, and morin at the same concentration on the mitochondrial membrane potential of HeLa cells. Our results revealed that paclitaxel and morin did not induce any significant changes in the membrane potential. In contrast, the kuwanon C-treated group displayed a noticeable decrease in membrane potential, as evidenced by increased green fluorescence and reduced red fluorescence (Figure 5B). Utilizing flow cytometry to detect the level of mitochondrial membrane potential, we observed that the kuwanon C-treated group exhibited a more pronounced decline in tumor cell membrane potential over time. Conversely, norartocarpetin, lacing an isopentenyl group, did not affect the membrane potential of mitochondria (Figure 5C,D).

### 2.6. Modulation of the Cell Cycle by Kuwanon C in HeLa Cells

In this study, we conducted flow cytometry analysis to investigate the effects of kuwanon C, kuwanon T, and morin on the cell cycle. The results revealed that as the concentration of kuwanon C increased, the proportion of HeLa cells in the G1/G0 and G2/M phases decreased, while the population of cells in the Sub G1 phase (indicative of DNA fragmentation) increased. Notably, there was no significant impact on DNA synthesis in the S phase. Conversely, morin did not exert any influence on the cell cycle. As for kuwanon T, an isoform differing from kuwanon C in a single isopentenyl position, it only affected the Sub G1 and G1/G0 phases, leaving the S and G2/M phases unaffected (Figure 6A,B). To further explore the underlying mechanisms, we assessed the expression of cycle-associated proteins at both the gene and protein levels. We specifically focused on SKP2 and PCNA, proteins associated with DNA synthesis and replication, as well as GADD34 and NOXA, proteins linked to apoptosis and endoplasmic reticulum stress. Our results demonstrated that the protein levels and gene expressions of PCNA, KIF20A, and SKP2 were down-regulated, while the expressions of CDKN1A and E2F1, related to cell cycle, apoptosis, and endoplasmic reticulum stress, were up-regulated (Figure 6C–F).

### 2.7. Molecular Targets of Kuwanon C

In order to find the target of kuwanon C in cells, we added the red fluorescent group CY3 to the hydroxyl position of kuwanon C (Figure 7A) and then treated cervical cancer HeLa cells with kuwanon C and morin labeled with the red fluorescent group CY3 (kuwanon C-CY3, morin-CY3). It was found that kuwanon C with isopentenyl groups could enrich onto the endoplasmic reticulum more rapidly than morin and CY3 and that kuwanon C was able to enter the nucleus and enrich onto the nucleolus as the treatment time increased. Only the CY3 group showed no significant enrichment effect (Figure 7B). No spontaneous red fluorescence appeared in the kuwanon C treatment group (Figure 7C). Meanwhile, after treatment, HeLa cells were simultaneously stained with mitochondrial green fluorescent probe and endoplasmic reticulum green fluorescent probe, and fluorescence microscopy photographic analysis showed that the position of red fluorescence of kuwanon C-CY3 in HeLa cells overlapped with the position of green fluorescence of mitochondria (Figure 7D,E) and endoplasmic reticulum (Figure 7H,I), and the results of the flow cytometry assay showed that the mitochondria (Figure 7F) and endoplasmic reticulum (Figure 7J), and the fluorescence signals were significantly enhanced to move to the right side in the extracted kuwanon C-CY3-treated group. Western blotting experiments confirmed that the extracted subcellular organelles were mitochondria (Figure 7G). Transmission electron microscopy also observed that the normal structure of mitochondria and the structure of the endoplasmic reticulum were severely disrupted in the kuwanon C-treated group (Figure 7K). For kuwanon C-treated HeLa cells without CY3 labeling, mitochondria were extracted and co-incubated with kuwanon C-CY3, and fluorescence microscopy photographs were taken to show that red fluorescent mitochondria were observed (Appendix A). Extracted mitochondria emitted red fluorescence after treatment of HeLa cells with red fluorescent group CY3-labeled morin (Appendix A). Kuwanon C-CY3-treated HeLa cells were labeled with a Golgi green fluorescent probe, which showed that the positions where the two markers were located overlapped (Appendix A).

Based on the structural features of kuwanon C, target site analysis at SwissTargetPredict website and Venn analysis showed that kuwanon C is likely to bind to cyclin kinase (CDK1) or cyclins (CCNB1, CCNB2) (Figure 7L,M). Mass spectrometry of the kuwanon C-CY3-bound protein bands identified the presence of CDK1 protein (Figure 7N), while molecular docking analysis of CDK1 and kuwanon C using the CB-DOCK2 website showed that CDK1 and kuwanon C were able to bind at a low binding energy (Vina = −10.1) (Figure 7O, Appendix A). The mass spectrometry identified that kuwanon C may bind to ATP synthase and cytochrome oxidase (Appendix A), and the transcriptome results of KEGG and GO enrichment analysis showed that genes related to TCA cycle, ROS formation, mitochondrial inner membrane proteins, oxidative phosphorylation, and nucleotide metabolism were significantly affected by kuwanon C treatment (Appendix A). The data from the transcriptome and STRING network analysis showed that the binding energy to the ATP synthase-related genes associated with ATP production was decreased in expression (Appendix A). 

## 3. Discussion

### 3.1. Disruption of Tumor Cell Energy Metabolism and Protein Synthesis by Kuwanon C

Mitochondria, commonly referred to as the ‘energy factory’ of the cell, play a vital role in regulating cellular activities, including ATP production, cellular respiration, oxidative phosphorylation, and apoptosis regulation [35,36]. Due to their multifaceted functions, mitochondria have become a prime target for the design of novel mitochondria-targeted anticancer drugs [37]. Efficient drug delivery to mitochondria enhances specificity and reduces drug toxicity [38]. Several clinically used anticancer drugs, such as molecularly targeted agents and chemotherapeutics, exploit mitochondria to induce the generation or burst of ROS with the intent of eliminating tumor cells [39]. Similarly, photodynamic therapy (PDT) and ionizing radiation therapy utilize ROS bursts to eliminate cancer cells [40]. The respiratory chain of mitochondria is the primary source of endogenous ROS. In our study, treatment of HeLa cells with kuwanon C resulted in a concentration-dependent increase in ROS levels. Interestingly, we observed that the number of isopentenyl groups in the compound correlated closely with ROS generation. When comparing kuwanon C (containing two isopentenyl groups) with albanin A (containing one isopentenyl group) and norartocarpetin (lacking isopentenyl groups) at a concentration of 40 μM for 24 h, kuwanon C exhibited significantly higher ROS generation, while the levels in the albanin A and norartocarpetin groups and the untreated group remained relatively low (Figure 3). ROS levels were also significantly higher in the kuwanon C-treated group compared to the paclitaxel-treated group, accompanied by a significant increase in intracellular calcium ion levels. Calcium ions and ROS are closely interrelated, mutually regulating each other, and mitochondria serve as the primary site of ROS generation [41]. Under physiological conditions, the body strictly regulates ROS homeostasis, wherein low levels stimulate cellular defense mechanisms through the upregulation of antioxidant enzymes to maintain health and longevity [42]. Conversely, high levels of ROS can impair proteins, lipids, and DNA, resulting in extensive oxidative stress and manual induction of mitochondrial DNA damage by ROS. Through JC-1 analysis, transcriptome analysis, and transmission electron microscopy experiments, our study revealed the downregulation of ATP generation-related enzymes and a significant decrease in mitochondrial membrane potential following kuwanon C treatment of HeLa cells. Furthermore, kuwanon C was found to target the mitochondrial and endoplasmic reticulum membranes. Transmission electron microscopy showed severe disruption of these structures in tumor cells, while transcriptome analysis indicated the downregulation of genes associated with ATP generation. Mass spectrometry analysis identified potential interactions between kuwanon C, ATP5F1A, and cytochrome oxidase. Collectively, these findings suggest that kuwanon C significantly inhibits ATP generation and protein synthesis in tumor cells, ultimately leading to cell death (Figure 8).

### 3.2. Impact of Isopentenyl Groups on the Antitumor Efficacy of Kuwanon C 

Isopentenyl groups exhibit diverse pharmaceutical activities, including anti-inflammatory, antibacterial, antiviral, and antitumor activities. For instance, geraniol with tandem isopentenyl group [33] and geranylgeraniol with four tandem isopentenyl groups have demonstrated antitumor effects [43]. Isopentenyl-substituted natural products serve as a valuable resource for innovative drugs due to their structural diversity and multifaceted pharmacological activities [31,44]. The incorporation of isopentenyl groups into natural product molecules not only enhances structural diversity but also improves affinity and bioavailability to drug targets by increasing lipophilicity, thereby enhancing drug formation properties. Isopentenyl groups can significantly disrupt cell membrane structure, increase membrane permeability, and enhance antimicrobial and antitumor capabilities [32,45,46]. In our experiments, it was also observed that the presence of isopentenyl groups affects mitochondrial membrane permeability, which in turn leads to a decrease in membrane potential (Figure 5). At the same time, the normal structure of mitochondria and endoplasmic reticulum is disrupted.

Numerous isopentenyl-containing compounds are naturally found in plants [34], including the Chinese medicinal plant mulberry, which harbors various isopentenyl compounds such as kuwanon series (A-W), rubraflavone C, albanin A, and artelasticin [45,47,48,49,50]. Isopentenyl groups can be connected individually at different positions on the compound framework or connected in tandem before linking to the compound’s skeleton [49,51,52]. The position and number of isopentenyl groups in these compounds influence their pharmaceutical properties [30,53,54]. Compounds containing isopentenyl groups exhibit greater pharmaceutical activity than those without, and compounds with a greater number of isopentenyl groups demonstrate higher activity. Our experiments yielded similar findings regarding the antitumor capability of kuwanon C. Moreover, we explored the influence of the number and position of the isopentenyl groups on the antitumor effect. Our experimental results revealed that kuwanon C, with isopentenyl groups at positions C3 and C8 in the flavonoid parent nucleus, exhibits stronger anti-proliferative and pro-apoptotic abilities compared to morin, which features a hydroxyl group at position C3. Additionally, we observed that altering the position of one isopentenyl group in kuwanon T and kuwanon C had minimal impact on the antitumor effect.

Previous studies have highlighted the susceptibility of single-target antitumor drugs to drug resistance upon prolonged administration, significantly undermining their effectiveness [55,56]. To address this challenge, the development of multi-target antitumor drugs offers advantages for long-term drug utilization [57,58]. With two isopentenyl groups, kuwanon C exhibits increased lipophilicity, which is beneficial for interactions with membrane organelles in tumor cells such as mitochondria and endoplasmic reticulum (Figure 8). Our experiments further support the multi-targeting effect of kuwanon C by demonstrating its strong binding affinity to both mitochondrial and endoplasmic reticulum membranes, providing compelling evidence.

### 3.3. Cellular Level Comparison of Kuwanon C with Established Clinical Anticancer Agents 

In this study, we conducted a cellular-level comparison of the clinically used antitumor drugs paclitaxel and cisplatin and observed that while paclitaxel exhibited antitumor effects at lower concentrations, its efficacy decreased as the concentration increased and was inferior to that of kuwanon C. Moreover, kuwanon C demonstrated superior antitumor effects compared to cisplatin. Additionally, kuwanon C displayed potent anti-proliferative effects on breast and brain tumor cells, with its efficacy surpassing that of paclitaxel as the concentration increased. The number and position of isopentenyl groups in kuwanon C also showed similar results to those observed in HeLa cells concerning anti-proliferative effects on breast cancer cells. Paclitaxel, derived from the Pacific yew tree and renowned for its remarkable antitumor effects and extensive utilization in various tumor treatments, has faced significant global demand. However, this high demand has resulted in extensive harvesting of the Pacific yew, endangering the species in the past. Efforts to address the demand by synthesizing paclitaxel have been challenging due to its highly complex structure, requiring numerous synthesis steps with low yields [59,60,61,62]. In contrast, kuwanon C is more easily accessible and abundant, as it can be found in the root bark, leaves, and fruits of the mulberry tree. Moreover, its structure is simpler than that of paclitaxel, making it more amenable to artificial synthesis. Therefore, with the completion of the mulberry genome sequencing [63] and mulberry metabolomics [47], the mulberry flavonoid kuwanon C has the potential to be developed into a safe and efficient antitumor drug.

## 4. Materials and Methods

### 4.1. Reagents

Kuwanon C, with a measured purity of 98.04% by HPLC, was obtained from Chengdu Yi Rui (Chengdu, China). Total RNA was extracted with TRIzol reagent (TIANGEN, Cat. #DP419, Beijing, China). The synthesis of cDNA utilized the PrimeScript RT Reagent Kit (TaKaRa, Cat. #RR047A, Dalian, China). SYBR Premix Ex Taq (TaKaRa, Cat. #RR420A, Dalian, China) and Hoechst 33258 Staining Solution (Sangon Biotech, Cat. #E607301, Shanghai, China) were used in the experiment. Fetal bovine serum (Sangon Biotech, Cat. #E600001, Shanghai, China), 0.25% Trypsin-EDTA (Gbico, Cat. #25200056, St Seaforth, ON, Canada), and penicillin/streptomycin (Gibco, Cat. #15140122, Bohemia, NY, USA) were used. Furthermore, BeyoClick™ EdU Cell Proliferation Kit with Alexa Fluor 488 (Beyotime Biotechnology, Cat. #C0071L, Shanghai, China) was employed. Crystal Violet Staining Solution (Beyotime Biotechnology, Cat. #C0121, China), DAPI Staining Solution (Beyotime Biotechnology, Cat. #C1006, Shanghai, China), and the BCA Protein Assay Kit (Beyotime Biotechnology, Cat. #P0012, Shanghai, China) were also used. Additionally, the experiments involved the use of the Mitochondrial Membrane Potential Assay Kit with TMRE (Beyotime Biotechnology, Cat. #C2001S, Shanghai, China), Reactive Oxygen Species Assay Kit (Beyotime Biotechnology, Cat. #S0033S, Shanghai, China), and the ER-Tracker Green Probe (Beyotime Biotechnology, Cat. #C1042S, Shanghai, China). Golgi-Tracker Green Probe (Beyotime Biotechnology, Cat. #C1045S, Shanghai, China), Cell Cycle and Apoptosis Analysis Kit (Beyotime Biotechnology, Cat. #C1052, Shanghai, China), Annexin V-FITC Apoptosis Detection Kit (Beyotime Biotechnology, Cat. #C1062S, Shanghai, China), and Fluo-4 Calcium Assay Kit (Beyotime Biotechnology, Cat. #S1061S, Shanghai, China) were employed in the experiment. The PrimeScript™ RT reagent Kit with gDNA Eraser (TaKaRa, Cat. #RR047A, Dalian, China) and Cell Plasma Membrane Staining Kit with DiO (Beyotime Biotechnology, Cat. #C1993S, Shanghai, China) were also utilized. Furthermore, the Immunol Staining Fix Solution (Beyotime Biotechnology, Cat. #P0098, Shanghai, China), Immunostaining Permeabilization Buffer with Saponin (Beyotime Biotechnology, Cat. #P0095, Shanghai, China), and One Step TUNEL Apoptosis Assay Kit (Beyotime Biotechnology, Cat. #C1090, Shanghai, China) were included in the experimental procedures. Additionally, the 2.5% Gluta Fixation Solution (Solarbio, Cat. #P1126, Beijing, China), Mitochondrial Extraction Kit (Solarbio, Cat. #SM0020, Beijing, China), and Endoplasmic Reticulum Extraction Kit (Solarbio, Cat. #EX2690, China) were used. Moreover, the CellTiter 96^®^ AQueous One Solution Cell Proliferation Assay (Promega, Cat. #G03581, Madison, WI, USA), Enhanced Mitochondrial Membrane Potential Assay Kit with JC-1 (Beyotime Biotechnology, Cat. #C2003S, Shanghai, China), and Dulbecco’s Modified Eagle’s Medium (DMEM) Low Glucose Culture Medium (Gibco, Cat. #C11885500BT, Shanghai, China) were used. 

### 4.2. Cell Lines and Cell Culture

The HeLa, MDA-MB-231, and LN229 cancer cells used in this study were derived from our laboratory’s preserved cells. These cell lines were cultured in Dulbecco’s Modified Eagle Medium (DMEM) (Gibco, Cat. #C111885500BT, Shanghai, China), which consists of 1g/L-D-Glucose, L-Glutamine, and 110 mg/L Sodium Pyruvate. The medium was supplemented with 10% fetal bovine serum (Sangon Biotech, Cat. #E600001, Shanghai, China) and 1% penicillin/streptomycin (Gibco, Cat. #15140122, Bohemia, NY, USA). T47D cells were cultured in RPMI 1640 supplemented with 10% fetal bovine serum and 1% penicillin/streptomycin. 

### 4.3. Cell Viability Assay

Cell viability was assayed using the CellTiter 96^®^ AQueous One Solution Cell Proliferation Assay (MTS) method, as described in previous studies. HeLa cells (3–5 × 10^4^) were seeded into 96-well plates and allowed to grow for 24 h. Subsequently, the cells were treated with different concentrations of kuwanon C, albanin A, morin, norartocarpetin, kuwanon T, paclitaxel, and cisplatin (ranging from 0 µM to 120 µM) for 12 h, 24 h, and 48 h. Following treatment, the medium was aspirated, and 100 µL of fresh medium was added to each well. Then, 20 µL of CellTiter 96^®^ AQueous One Solution Reagent was added to the wells, and the plates were incubated at 37 °C, 5% CO_2_ for 2 h. The optical densities were measured at 490 nm using a microplate reader (Synergy H1, Burlington, Vermont, USA). The absorbance value of the control group was considered as 100% cell viability. All experiments were independently repeated three times.

### 4.4. Wound Healing Assays

A wound healing test was performed to evaluate the migratory capacity of the cells. HeLa cells were inoculated in 6-well cell culture plates and allowed to reach approximately 100% confluency. Subsequently, cell scratching was performed. Three replicates were prepared, and the cells were treated with 20 μM and 40 μM concentrations of kuwanon C, albanin A, and norartocarpetin. The cells were incubated at 37 °C with 5% CO_2_ for 24 h. After incubation, cell migration was assessed using an Olympus IX73 microscope (Olympus Corporation, Tokyo, Japan). The gap in the scratch was visually examined and photographed, and the area of the scratch gap was quantified using Image J (version, 1.X) software for migration rate analysis.

### 4.5. Colony Formation Assay

HeLa cells were dissociated from cell culture flasks using trypsin and then seeded at a density of 700–1000 cells per well in 6-well cell culture plates. The culture medium contained 1% penicillin/streptomycin and 10% fetal bovine serum (FBS). The cells were incubated at 37 °C with 5% CO_2_ and 95% humidity for 48 h. Subsequently, the cells were treated with different concentrations (0 μM, 1 μM, 2 μM, 5 μM, 10 μM, 20 μM) of kuwanon C, albanin A, and morin for 96 h. After treatment, they were washed once with PBS, fixed with 4% paraformaldehyde for 15 min, and then stained with 1% crystalline violet at 37 °C for 30 min. Following staining, the cells were washed 3–5 times with PBS and air-dried, and the colonies were analyzed by capturing images of each well in the 6-well plate using a digital camera and microscope. 

### 4.6. Transmission Electron Microscopy

HeLa cells were cultured in 6-well cell culture plates until they reached approximately 80% confluency. At this point, the cells were treated with 60 μM and 80 μM concentrations of kuwanon C for 24 h and 48 h. Following treatment, cells from the three wells were collected into 1.5 mL centrifuge tubes and centrifuged at 2000 rpm for 10 min. The supernatant was carefully removed, and the cellular precipitates were collected. Then, 1 mL of electron microscopy fixative was added to each sample, and the samples were briefly stored in a refrigerator. Subsequently, the samples were processed according to the transmission electron microscopy sample processing procedure. Finally, the samples were photographed and analyzed using a transmission electron microscope (Hitachi High-Tech Corporation, HT7800, Tokyo, Japan).

### 4.7. 5-Ethynyl-2′-Deoxyuridine (EdU) Staining

HeLa cells were enzymatically detached using trypsin and subsequently seeded into 12-well cell culture plates. After incubation, when the cell density reached approximately 80%, the cells were subjected to different concentrations of kuwanon C (0 μM as control, 25 μM, and 50 μM). Additionally, cells were treated with the same concentration (50 μM) of kuwanon C, albanin A, and norartocarpetin for a duration of 24 h. Following the treatment period, the cells were assayed using BeyoClick™ EdU-488 Cell Proliferation Assay Kit according to the provided protocol. This was followed by nuclei staining using Hoechst 3342 dye. Subsequently, the cells were washed, and an anti-fluorescence bursting agent was added. Finally, fluorescence microscopy was used for photographic analysis of the samples.

### 4.8. Kuwanon C and Morin Red Fluorescent Group CY3 Labeling

Kuwanon C and morin were sent to Xi’an Qiyue Biotechnology Co., Ltd (Xi’an, China). for CY3 labeling of the red fluorescent group. The successful labeling of kuwanon C and morin with CY3 enabled the tracking of kuwanon C binding sites in the cells during subsequent experiments.

### 4.9. Bioinformatic Analysis and Mass Spectrometric Characterization of Target Molecules for Kuwanon C Action

To explore the molecular characteristics of kuwanon C, albanin A, norcartocarpetin, and morin, their chemical structures were analyzed using the SwissTargetPrediction website (http://www.swisstargetprediction.ch/result.php?job=417850766&organism=Homosapiens, accessed on 12 April 2022). Additionally, a targeted analysis of the binding sites was conducted. HeLa cells were treated with CY3-labeled kuwanon C for 24 h, and the cells were collected at the end of the treatment to extract proteins. Non-denaturing page electrophoresis was then performed, and upon completion, the gel was fluorescently excited, revealing red protein bands. Subsequently, the red fluorescent protein bands were excised from the gel and sent to Applied Protein Technology for identification and analysis of the protein profiles.

### 4.10. Molecular Docking

We utilized the molecular docking platform CB-DOCK2 (https://cadd.labshare.cn/cb-dock2/php/index.php, accessed on 15 April 2022) [64] to perform docking simulations of kuwanon C with its predicted target proteins. The receptor (target protein) and ligand (kuwanon C) information were imported separately into the platform. The default setting of 5 active cavities for docking was selected, and automatic blind docking was conducted. Post-docking, active cavities with Vina scores lower than −7 Kcal/mol were considered to indicate strong binding activity between the ligand and receptor [65]. These selected active cavities were then analyzed to identify the specific residues involved in the interaction between the ligand and the receptor.

### 4.11. Mitochondrial Membrane Potential Measurement

HeLa cells were cultured in 6-well and 12-well cell plates until they reached approximately 80% confluency. HeLa cells were then treated with different concentrations of kuwanon C (0 μg/mL, 20 μg/mL, 30 μg/mL, and 40 μg/mL) for 8 h, and changes in mitochondrial membrane potential were detected using Beyotime TMRE staining. Fluorescence microscopy was employed to capture images and detect alterations in mitochondrial membrane potential. Additionally, HeLa cells were exposed to kuwanon C, morin, and paclitaxel at a concentration of 30 μg/mL for 24 h. The cells were then stained with Biotin JC-1 and imaged under a fluorescence microscope to evaluate changes in mitochondrial membrane potential. For further investigation, cells from the 6-well plate were treated with 60 μM concentrations of kuwanon C, albanin A, and norartocarpetin for 12 h and 24 h. Staining with Beyotime JC-1 was carried out for 12 h and 24 h, followed by fluorescence microscopy to assess the changes in mitochondrial membrane potential. Flow cytometer (URIT Flow Cytometer, BF-700, URIT Medical Electronic Co., Ltd., Guilin, China) utilizing Beyotime JC-1 staining was employed to analyze changes in mitochondrial membrane potential.

### 4.12. Mitochondrial Staining

HeLa cells were inoculated into 12-well cell culture plates and incubated until they reached approximately 80% confluency. The cells were treated with 10 μM kuwanon C-CY3 for 1 h. Following this treatment, the cells were stained with a Mito-Tracker Green fluorescent probe. The stained cells were then subjected to fluorescence microscopy analysis using an Olympus IX73 inverted microscope (Olympus Corporation, Tokyo, Japan). Images were captured and analyzed to investigate the fluorescence patterns and distribution of the stained mitochondria within the cells.

### 4.13. Endoplasmic Reticulum Staining

HeLa cells were inoculated into 12-well cell culture plates and incubated until they reached approximately 80% confluency. The cells were treated with 10 μM kuwanon C-CY3 for 1 h. To visualize the endoplasmic reticulum (ER), ER-Tracker Green was diluted at a ratio of 1:1000 and added to the cells. The cells were then incubated with the ER-Tracker Green solution at 37 °C for 30 min. Afterward, the cells were washed with PBS to remove any excess dye. The samples were observed and photographed using a fluorescence microscope to examine the distribution and fluorescence of the endoplasmic reticulum.

### 4.14. HeLa Cell Mitochondria Extraction and Analysis

HeLa cells were cultured in 6-well cell culture plates, and once the cell growth reached approximately 80%, the cells were treated with kuwanon C as well as CY3-labeled kuwanon C (kuwanon C-CY3). After 24 h of treatment, the cells were collected and extracted according to the procedure of the Solarbio Mitochondrial Extraction Kit. For the group treated solely with kuwanon C, 10 μL of the extracted mitochondrial solution was obtained and mixed with kuwanon C-CY3 for a 30 min incubation period. After incubation, the mitochondria were centrifuged, washed with PBS, resuspended in PBS, and, subsequently, 2 μL of the resuspended solution was collected for fluorescence microscopy photography. In the case of the groups treated with both kuwanon C and kuwanon C-CY3, 2 μL of the extracted mitochondria was placed onto a carrier slide, covered with a coverslip, and then observed and recorded under a sunlight microscope. The remaining extracted mitochondria were utilized for flow cytometry assay (Wellgrow EasyCell Flow Cytometer, Shenzhen Wellgrow Technology Ltd., Shenzhen, China) as well as protein extraction. The extracted mitochondria were further characterized using marker proteins specific to mitochondria.

### 4.15. ROS Assays

HeLa cells were inoculated into both 6-well and 12-well plates. Once the cell growth reached approximately 80%, the cells were treated with different concentrations of kuwanon C, albanin A, norartocarpetin, and paclitaxel, as well as antioxidant and N-acetylcysteine for 3 h, 6 h, 12 h, and 24 h, respectively. After the respective treatment periods, the cells were analyzed using the Beyotime ROS Detection Kit (Beyotime, Cat. #S00033s, Shanghai, China). Fluorescence microscopy was used to capture images, while flow cytometry (Wellgrow EasyCell Flow Cytometer, Shenzhen Wellgrow Technology Ltd., Shenzhen, China) was utilized to measure the levels of ROS.

### 4.16. Transcriptome Analysis

HeLa cells were inoculated into 6-well cell culture plates and incubated until reaching an approximate cell growth density of 80%. The cells were treated with 30 μg/mL of kuwanon C for 24 h at 0 μg/mL as a treatment control for 24 h. After treatment, the cells were collected and stored at −80 °C. The cell samples were sent to Applied Protein Technology for transcriptome sequencing and analysis.

### 4.17. Cellular Calcium Content Analysis

HeLa cells were cultured in 12-well cell culture plates until they reached 80% growth density. The cells were subjected to a 24 h treatment with kuwanon C, albanin A, and norartocarpetin, all at a concentration of 60 μM kuwanon C. To observe the intracellular calcium ion level, the Fluo-4 Calcium Ion Detection Kit was utilized. Each well was supplied with 400 μL of Flou-4 Calcium Ion Staining Solution, followed by a 30 min incubation at 37 °C. Afterward, the cells were analyzed and photographed under a fluorescence microscope (Olympus IX73 microscope, Olympus Corporation, Tokyo, Japan).

### 4.18. Gene Expression Analyses

HeLa cells were inoculated into 6-well plates and allowed to reach the desired density. The cells were then treated with a consistent concentration (60 μM) of kuwanon C, albanin A, and norartocarpetin for 12 h, 24 h, and 48 h. The cells were collected, and cDNA was extracted using the TIANGEN total RNAsimple Extraction Kit (TIANGEN, Cat. #DP419, Beijing, China). The cDNA was generated using Takara PrimeScript™ RT reagent Kit (TaKaRa, Cat. #RR047A, Dalian, China) with gDNA Eraser, and the resulting samples were stored at −80 °C. For fluorescence quantification, the TIANGEN SuperReal Premix Plus (TIANGEN, Cat. #FP205, Beijing, China) was used as the quantification reagent. The primers for gene quantification were designed with Primer Premier 5 (Version 5.0, PREMIER Biosoft International, Palo Alto, Santa Clara, CA, USA) and then were synthesized by Tsingke Biotechnology Co. Ltd. (Chongqing, China).

### 4.19. Western Blotting Analyses

HeLa was inoculated into 6-well cell culture plates and treated with various compounds, including kuwanon C, albanin A, norartcarpetin, kuwanon T, morin, and paclitaxel for 24 h. Proteins were extracted using RIPA protein lysis buffer containing 1% PMSF, and the total protein content was assessed using BCA. Proteins were then separated by SDS-PAGE and transferred onto PVDF membranes. The membranes were blocked with 5% skimmed milk at 4 °C overnight. The following day, after rinsing with TBST buffer, the membrane was incubated with primary antibody for 2 h at 37 °C. The membranes were washed three times with TBST for 10 min each and incubated with secondary antibodies for 40–60 min. Finally, the PVDF membranes were washed again with TBST, and the protein signal was detected using the Beyotime ECL Plus ECL reagent (Beyotime, Cat. #P0018S, Shanghai, China). The primary antibodies used included SKP2 (CST, Cat. #2652S, Danvers, MA, USA) (1:1000), NOXA (CST, Cat. #14766S, Danvers, MA, USA) (1:1000), GADD34 (CST, Cat. #41222S, Danvers, MA, USA) (1:1000), and GAPDH (CST, Cat. #2118S, Danvers, MA, USA) (1:1000).

### 4.20. Cell Cycle Assay

The cell cycle assay comprised three steps: (1) Cell preparation: Six groups were established for the assay, including a control group, as well as groups treated with 30 μM kuwanon C, 60 μM kuwanon C, 50 μM kuwanon C, 50 μM kuwanon T, and 50 μM morin. After a 24 h treatment, single-cell suspensions were obtained by trypsin digestion. The cells were then centrifuged at 1000× *g* for 5 min, and the supernatants were discarded. The cell precipitates were collected and resuspended with 1 mL of pre-cooled PBS. Following another centrifugation step at 1000× *g* for 5 min, the cell pellets were washed once more with 1 mL of pre-cooled PBS and collected. (2) Cell fixation: The collected cell pellets from each group were fixed overnight at 4 °C by adding 1 mL of pre-cooled 70% alcohol at −20 °C. The cells were centrifuged at 1000× *g* for 5 min, the supernatant was discarded, and the cell pellets were collected. Resuspension of the cells in 1 mL of pre-cooled PBS was followed by another round of centrifugation at 1000× *g* for 5 min. After discarding the supernatant, the cell pellets were collected. (3) PI staining: The cells were then gently suspended in 500 μL PI/RNase A staining solution according to the instructions provided with the cell cycle assay kit. The stained cells were analyzed using a Beckman CYTOFLEX flow cytometer (Beckman, IN, USA), with 10,000 cells collected in each group.

### 4.21. Statistical Analysis

The data were expressed as mean ± SEM. Group comparisons were conducted using either two-way analysis of variance (ANOVA) with the Newman–Keuls multiple comparisons test or one-way ANOVA followed by the Bonferroni multiple comparisons test or the uncorrected Fisher’s LSD test. For experiments with only two groups, a *t*-test was applied for comparison. The GraphPad Prism 9 software package was used to analyze all the data.

## 5. Conclusions

Our study demonstrates that kuwanon C, an isopentenyl flavonoid obtained from the root bark of mulberry, exerts potent antitumor effects. Through its interaction with the mitochondrial and endoplasmic reticulum membranes, kuwanon C induces a significant production of ROS, leading to the disruption of their normal structure. This disruption alters ATP production, impairs protein synthesis and processing, and inhibits cell cycle progression. Moreover, kuwanon C stimulates apoptotic signaling pathways, ultimately resulting in the death of HeLa tumor cells. Furthermore, we observed anti-proliferative and pro-apoptotic effects of kuwanon C on breast cancer and brain tumor cells.

## Figures and Tables

**Figure 1 ijms-25-08293-f001:**
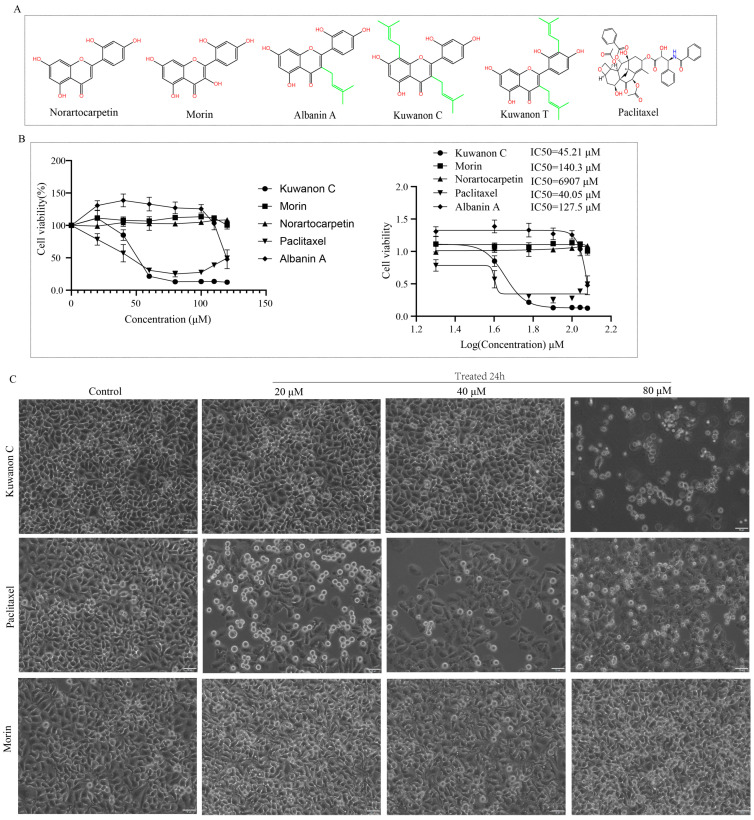
Inhibition of HeLa cell viability by kuwanon C. (**A**) Molecular structure of norartocarpetin, morin, albanin A, kuwanon C, kuwanon T, and paclitaxel. (**B**) Effects of increasing concentrations of kuwanon C, morin, norartocarpetin, paclitaxel, and albanin A on HeLa cell viability as determined by MTS assay. (**C**) Comparison of the cytotoxicity of kuwanon C, morin, and paclitaxel on HeLa cells. Each bar represents the mean ± SD of three independent experiments; scale bar = 50 μm.

**Figure 2 ijms-25-08293-f002:**
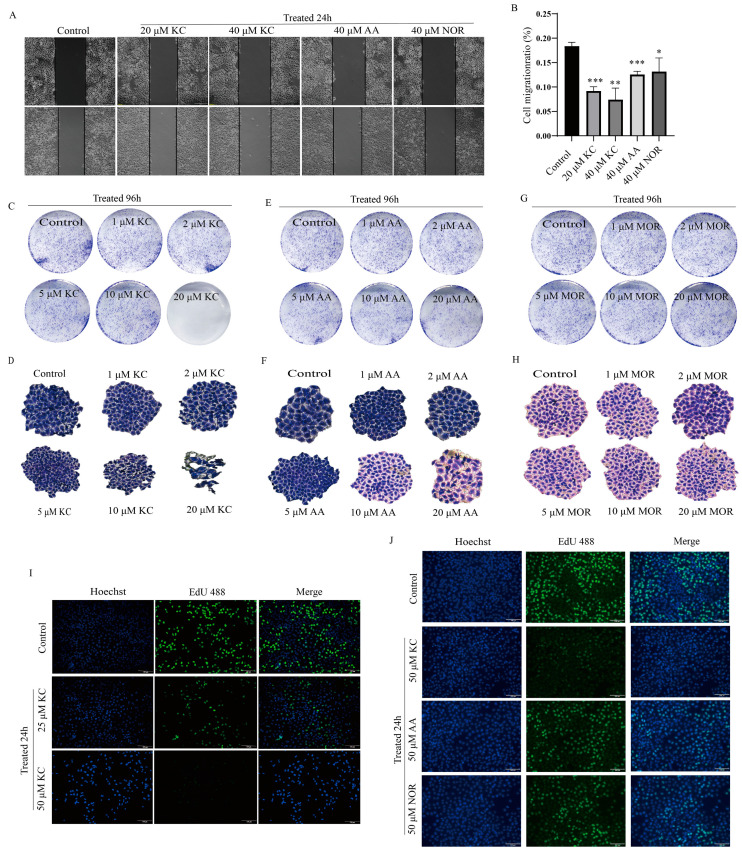
Suppression of HeLa cell migration and proliferation by kuwanon C. (**A**,**B**) Different concentrations of kuwanon C (with 2 isopentenyl groups) and albanin A (with 1 isopentenyl group) and norartocarpetin (containing 0 isopentenyl groups) were examined for the assessment of the migratory ability of HeLa cells by cell scratch assay. Scale bar = 200 μm. (**C**–**H**) Clone formation ability of kuwanon C, albanin A, and morin on HeLa cells was assessed by plate clone formation assay. (**I**,**J**) EdU 488 staining method was used to detect the effect of different concentrations (0–50 μM) of kuwanon C on the proliferative capacity of tumor cells. Scale bar = 100 μm Each bar represents the mean ± SD of three independent experiments. Statistical significance is indicated by * *p* < 0.05, ** *p* < 0.01, *** *p* < 0.001.

**Figure 3 ijms-25-08293-f003:**
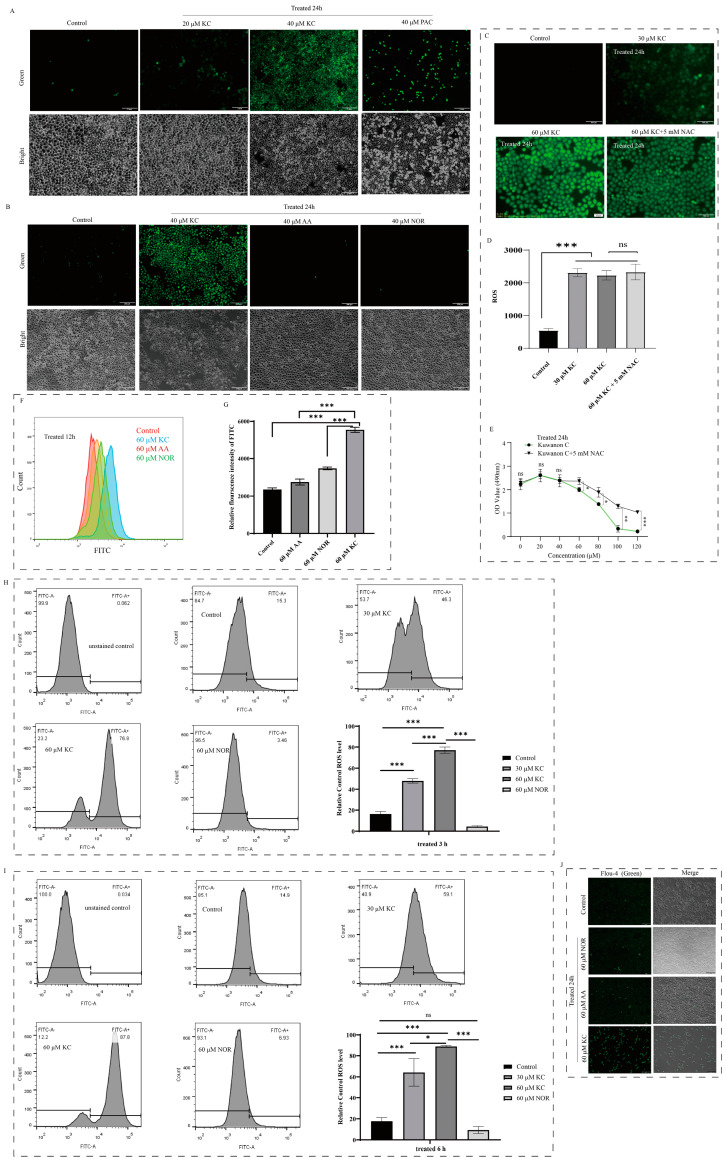
Induction of reactive oxygen species generation in HeLa cells by kuwanon C. (**A**,**B**) HeLa cells were treated with kuwanon C (KC), albanin A (AA), norartocarpetin (NOR), and paclitaxel (PAC) for 24 h. Subsequently, the cells were exposed to a ROS detection reagent, and the levels of ROS were measured using fluorescence microscopy. (**C**,**D**) Treatment of HeLa cells with 60 μM kuwanon C in the presence of 5 mM of the antioxidant N-Acetylcysteine (NAC). (**E**) Cell viability assay, effect of antioxidant NAC on cell viability after kuwanon C treatment of HeLa. (**F**,**G**) Flavonoid compounds with different numbers of isopentenyl groups, kuwanon C, AA, and NOR, were treated with HeLa cells for 12 h and intracellular reactive oxygen species levels were measured by flow cytometry. (**H**,**I**) The HeLa cells were treated with different concentrations of kuwanon C (0, 30 μM, 60 μM) and 60 μM NOR for 3 h and 6 h, respectively, and then the intracellular levels of reactive oxygen species were detected by flow cytometry. (**J**) Intracellular calcium imaging. Green fluorescence serves as an indicator of increased intracellular calcium ion concentration. Scale bar = 200 μm. Each bar represents the mean ± SD of three independent experiments. Statistical significance is indicated by * *p* < 0.05, *** *p* < 0.001 and ns: not significant.

**Figure 4 ijms-25-08293-f004:**
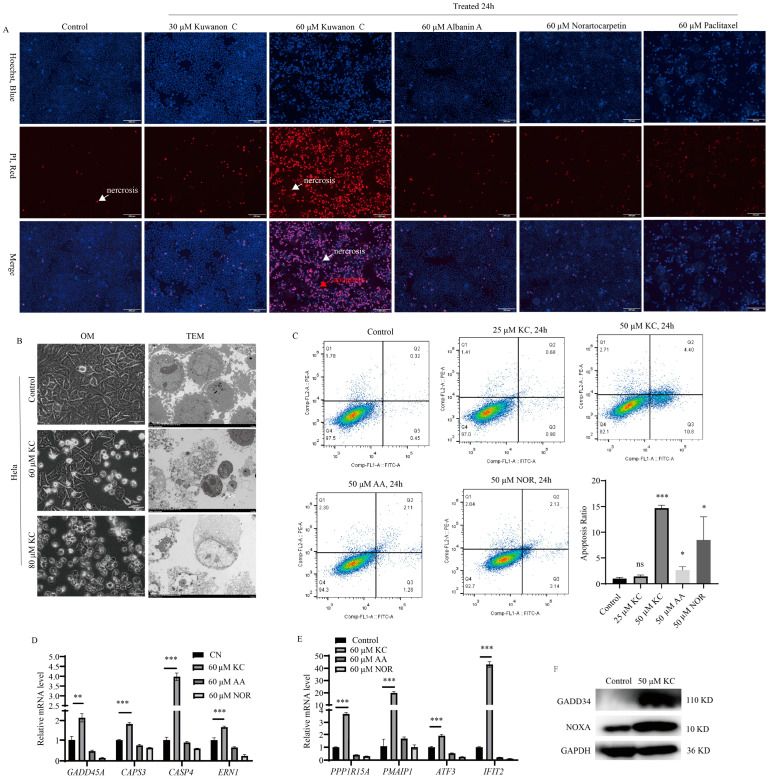
Induction of apoptosis in HeLa cells by kuwanon C. (**A**) Apoptosis and necrosis staining was used to detect the apoptosis and necrosis level of HeLa cells treated with 0 μM (control), 30 μM, 60 μM of kuwanon C, albanin A, norartocarpetin, and paclitaxel for 24 h. (Red arrows indicate apoptotic cells, and white arrows indicate necrotic cells). Scale bar = 200 μm. (**B**) The impact of kuwanon C on the microstructure of HeLa cells. HeLa cells were treated with 0 μM, 60 μM, and 80 μM kuwanon C and examined using transmission electron microscopy. Scale bar = 10 μm. (**C**) Flow cytometry was performed to detect the apoptosis level of HeLa cells after treatment with different concentrations (control, 25 μM, 50 μM) of kuwanon C, albanin A (50 μM), and norartocarpetin (50 μM) for 24 h. (**D**,**E**) The expression levels of apoptosis-related genes (*GADD45A*, *Casp3*, *Casp4*, *ERN1*, *PPP1R15A*, *PMAIP1*, *ATF3*, and *IFIT2*) were quantified using (RT-qPCR). (**F**) Western blotting detection of protein levels of apoptosis-related proteins GADD34, NOXA. Each bar represents the mean ± SD of three independent experiments. Statistical significance is indicated by * *p* < 0.05, ** *p* < 0.01, *** *p* < 0.001 and ns: not significant.

**Figure 5 ijms-25-08293-f005:**
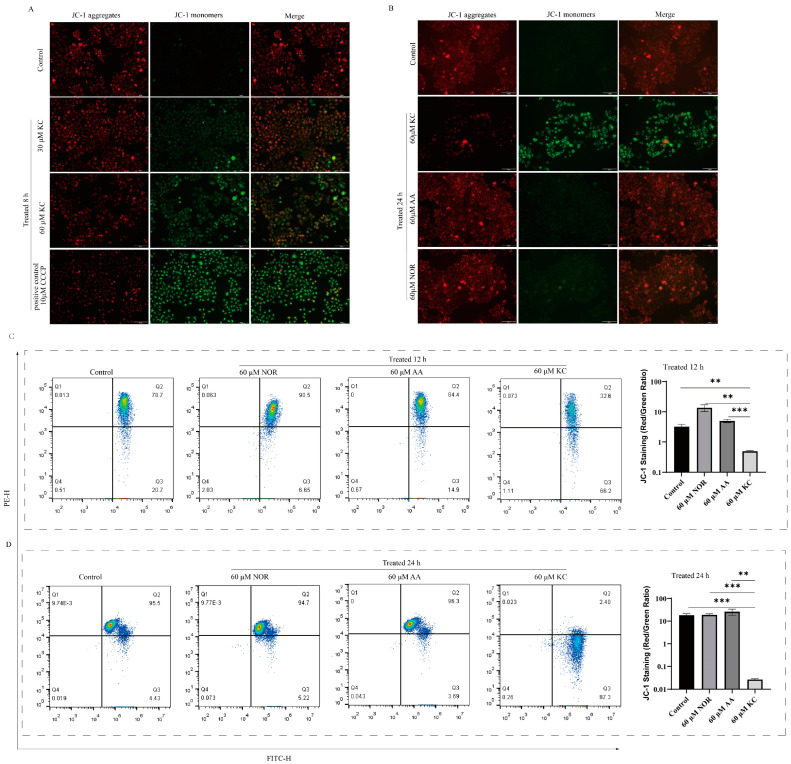
Kuwanon C decreases mitochondrial membrane potential. (**A**) Changes in cell membrane potential were assessed using JC-1 staining, where strong red fluorescence indicated high membrane potential and vice versa. Scale bar = 50 μm. (**B**) The impact of kuwanon C (KC), albanin A (AA), and norartocarpetin (NOR) on HeLa cell membrane potential was detected using JC-1 staining. HeLa cells were treated with the same concentrations (60 μM) of each compound for 24 h. Scale bar = 50 μm. (**C**,**D**) Flow cytometry was performed to measure the mitochondrial membrane potential levels in HeLa cells treated with kuwanon C (KC), albanin A (AA), and norartocarpetin (NOR) for 12 and 24 h, respectively. Each bar represents the mean ± SD of three independent experiments. Statistical significance is indicated by ** *p* < 0.01, *** *p* < 0.001.

**Figure 6 ijms-25-08293-f006:**
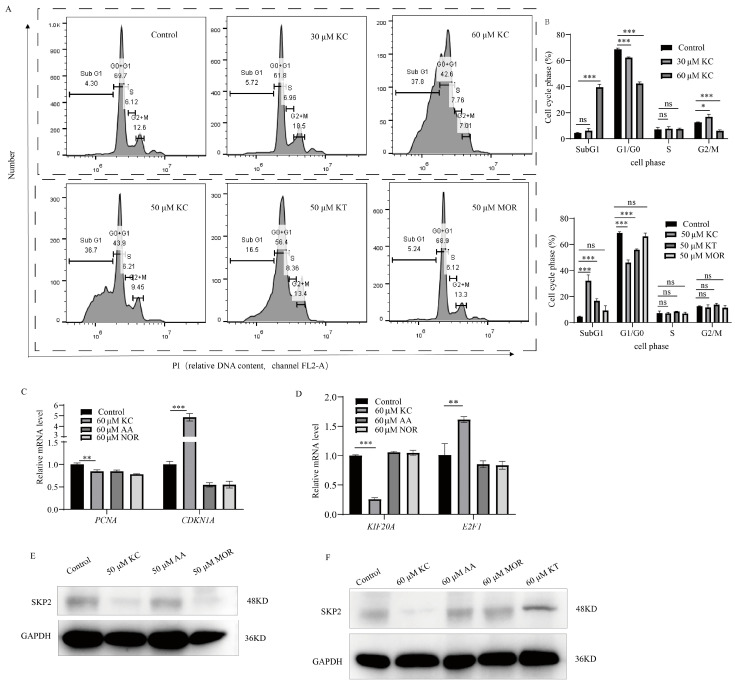
Modulation of the cell cycle by kuwanon C in HeLa cells. (**A**,**B**) Flow cytometry was performed to investigate the impact of kuwanon C (KC: 30 μM, 50 μM, and 60 μM), kuwanon T (KT, 50 μM), and morin (MOR, 50 μM) on the cell cycle of HeLa cells following a 24 h treatment. (**C**,**D**) Analysis of the expression levels of cell cycle-related genes (*PCNA*, *CDKN1A*, *KIF20A*, *E2F1*) by differences in the number of isopentenyl groups of flavonoid compounds (kuwanon C, albanin A, norartocarpetin). (**E**,**F**) The expression levels of cycle-related proteins were evaluated by Western blotting, with GAPDH serving as the internal reference. Specifically, the protein levels of SKP2, which are involved in DNA synthesis and replication, were investigated. Each bar represents the mean ± SD of three independent experiments. Statistical significance is indicated by * *p* < 0.05, ** *p* < 0.01, *** *p* < 0.001 and ns: not significant.

**Figure 7 ijms-25-08293-f007:**
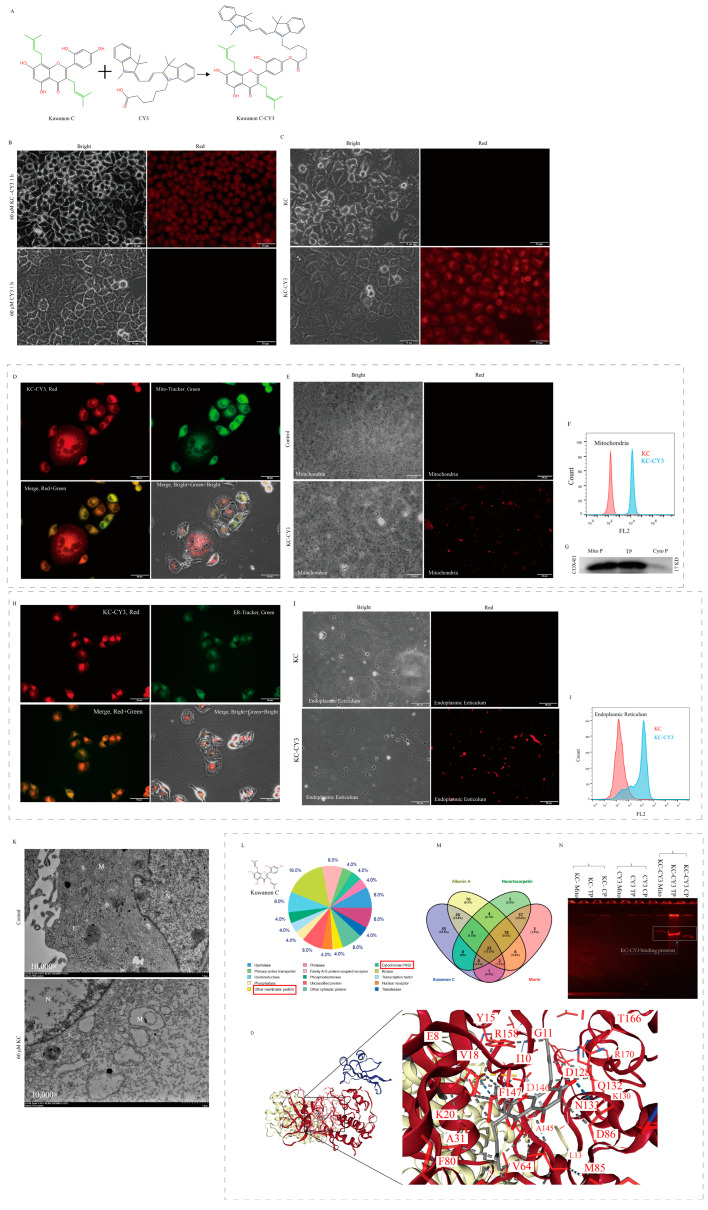
Molecular targets of kuwanon C. (**A**) Kuwanon C was labeled with the red fluorescent moiety CY3. (**B**,**C**) Upon treatment of HeLa cells with kuwanon C-CY3, CY3, and kuwanon C, only the cells in the kuwanon C-CY3 treatment group emitted red fluorescence. Scale bar = 50 μm. (**D**,**H**) HeLa cells were treated with CY3-labeled kuwanon C for 24 h, followed by labeling with green fluorescent probes targeting mitochondria and endoplasmic reticulum. Fluorescence microscopy was utilized to visualize the binding site of kuwanon C. (**E**,**I**) The extracted organelles of mitochondria and endoplasmic reticulum were observed and characterized through fluorescence microscope examination. (**F**,**J**) Flow cytometry analysis was conducted to assess the extracted organelles from the kuwanon C-CY3 and kuwanon C-treated groups. (**G**) Western blotting was performed to further verify the presence of kuwanon C in the extracted organelles. (**K**) The status of the endoplasmic reticulum and mitochondria in HeLa cells was observed using transmission electron microscopy (Scale bar = 1 μm; Abbreviations used: endoplasmic reticulum: E, mitochondria: M, nucleus: N). (**L**) The prediction of kuwanon C target proteins. (**M**) Venn analysis of the target proteins of kuwanon C, albanin A, norartocarpetin, and morin. (The proteins shown in red box are a family of membrane proteins to which Kuwanon C may bind.) (**N**) Non-denaturing PAGE electrophoresis experiments were conducted using extracted organelles and total proteins from HeLa cells treated with kuwanon C-CY3. (**O**) Molecular docking analysis of kuwanon C with target cycle-associated protein CDK1 (Abbreviations used: kuwanon C: KC; albanin A: AA; norartocarpetin: NOR; cyclin-dependent kinase 1: CDK1; mitochondria: Mito; total protein: TP; cytoplasm protein: CP).

**Figure 8 ijms-25-08293-f008:**
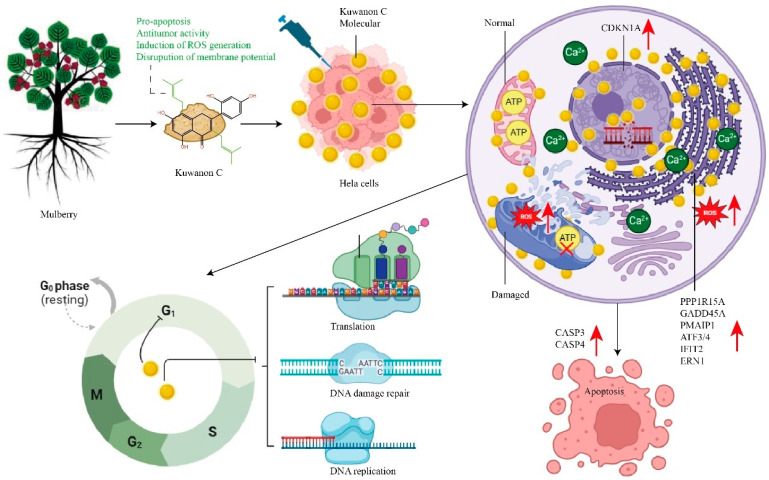
The schematic illustrated the mechanism of antitumor action of kuwanon C. After the treatment of HeLa cells with kuwanon C extracted from the root bark of mulberry, kuwanon C penetrated the cell cytoplasm and accumulated in the membranes of the endoplasmic reticulum and mitochondria. This accumulation resulted in the induction of endoplasmic reticulum stress, damage to the mitochondrial structure, and an elevated production of reactive oxygen species (ROS). As a consequence, the expression of pro-apoptotic genes was activated, impeding the advancement of the cell cycle and ultimately leading to the demise of HeLa cells. Red forks indicate inhibited ATP production, red arrows indicate increased gene expression levels and increased ROS production, and yellow balls indicate kuwanon C molecules.

## Data Availability

The original contributions presented in the study are included in the article.

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
