# Peer review of "Kuwanon C Inhibits Tumor Cell Proliferation and Induces Apoptosis by Targeting Mitochondria and Endoplasmic Reticulum"

_ijms, 2024, doi:10.3390/ijms25158293_

Round 1

Reviewer 1 Report

Comments and Suggestions for Authors

The topic of the paper is very interesting showing an amazing anticancer property of a natural compound. Using natural products in oncology  offers a lot of benefits including no toxicity for non-tumor cells.

To hope for the clinic utilization of Kuwanon C, the authors should investigate its safety for normal cells, for examples fibroblasts or endothelial cells.

All the images should be revised adding a few captions to make comprehension more immediate and in figure 5 A,B concentrations should be expressed as molarity rather than in ug/ml.

Figure 7 should be enlarged to be able to read the captions well

Author Response

Comments 1: To hope for the clinic utilization of Kuwanon C, the authors should investigate its

safety for normal cells, for example, fibroblasts or endothelial cells.

Response 1: We express our profound gratitude to the reviewers for their invaluable review

comments. we examined the impact of Kuwanon C on human umbilical vein vascular endothelial cells

(HUVEC). Regrettably, the findings indicated that Kuwanon C also hindered the growth of HUVEC

cells, comparable to its effects on tumor cells (Supplementary Fig. S5). While we did not discover

Kuwanon C specifically targets tumor cells, and our findings demonstrated that at the cellular level of

cervical and breast cancer, the anti-tumor impact of high concentrations of Kuwanon C surpassed that

of paclitaxel at the same concentration. In the future, it is possible to combine Kuwanon C with nano-drug delivery system to specifically transport Kuwanon C to tumor tissues and cells, resulting in

in focused anti-tumor therapy and minimized side effects. Extensive research is required to achieve

this in the future. Thank you for pointing this out. We agree with this comment. The results of

the experiments on the effect of Kuwanon C on normal cells can be found in the Supplementary

Material, Figure S5. (See supplementary material, page 5, Lines 40 and 43.)

Comments 2: All the images should be revised, adding a few captions to make comprehension more

immediate and in figure 5 A, B concentrations should be expressed as molarity rather than in ug/ml.

Response 2: Agree. Following the insightful recommendations from the reviewers, we have finished

the process of identifying all image captions. Furthermore, the concentration unit (μg/mL) in Figures

5A and 5B have been changed to molar concentration, aligning it with the concentration unit used in

the full manuscript. It should be pointed out to the reviewers that this unit of concentration was used

only for the early study of Kuwanon C. However, after the subsequent study found that the

isopentenyl structure of Kuwanon C plays a key role in its effect on tumor cells and other substances

with structural differences in isopentenyl structure were introduced, and it was not appropriate to use

the unit of μg/mL again. In conjunction with the feedback from another reviewer, we conducted a

re-examination of this section of the mitochondrial membrane potential experiment. Reviewers are

invited to view the revised image annotations in Figures 1–8, and the concentration unit

revisions in Figures 5A and 5B in Figure 5. (See page 4, line 115; page 6, line 140; page 8, line 176.)

page 10, line 214; page 12, line 244; page 13, line 270; page 15, line 323.

Comments 3: [Figure 7 should be enlarged to be able to read the captions well.]

Response 3: We have rearranged Figure 7 and also enlarged the image for better understanding of

the experimental results. Reviewers are invited to review it again. Thank you for pointing this out.

out. We agree with this comment. (See page 15, line 323.)

Reviewer 2 Report

Comments and Suggestions for Authors

The paper focuses on basic research on Kuwanon C and its anti-cancer properties. Several critical issues need to be addressed to meet the journal's standards and improve the overall quality of the manuscript.

1. Journal Guidelines: The authors should thoroughly read the instructions provided by the journal MDPI International Journal of Molecular Sciences and revise the manuscript accordingly.

 2. Structure of the Manuscript: Information in the introduction and discussion/conclusion sections needs to be properly separated. The introduction currently contains too many specific details about the study results, which should be reserved for the results and discussion sections.

 3. Selective Toxicity: There is a lack of experiments on healthy cell lines to determine the selectivity of Kuwanon C and justify the relevance of conducting such studies. Including these would provide a more comprehensive understanding of the compound's safety and potential therapeutic window.

 4. Figure 1 Adjustments: In Figure 1, the graphs labeled "B" should have the y-axis labeled as "cell viability in %" instead of "OD." Additionally, IC50 values should be calculated and presented in the text or a table for all tested cell lines. Control compounds (Morin, Paclitaxel, Albanin A) should also be tested using the MTS assay, and their values reported.

 5. HeLa Cell Line Justification: The rationale for selecting the HeLa cell line for further studies is missing. This information should be included to clarify why this particular cell line was chosen.

6. Image Quality: Figures 2-7 are inferior quality, making it difficult to review the results. Original images must be provided in a single zip archive as supplementary material files at a sufficiently high resolution (a minimum of 1000 pixels in width/height, or a resolution of 300 dpi or higher).

7. Scale Bars: Figures 2 and 5 are missing scale bars.

8. Migration Assay: The migration experiment is flawed because Kuwanon C at 20 µM inhibited HeLa cell proliferation after 24 hours of incubation. Therefore, the observed effects in the scratch assay are likely due to inhibited proliferation rather than migration. The experiment should be redone using a DNA synthesis inhibitor like Mitomycin C, which is the standard practice.

 9. Terminology: The term "tumorigenic capacity" is not appropriate for in vitro studies. The manuscript should use more accurate terminology to describe the findings.

10. Western Blot Figures: For the membranes shown in Figure 4F, full-length images of the blots should be provided.

11. Microscopy Annotations: Apoptotic/necrotic cells, etc., should be indicated with arrows in the images (Figure 4F) and described in the figure legend.

 12. Formatting Issues: Section 3.3 is entirely written in italics and should be corrected to match the formatting of the rest of the manuscript.

Author Response

Comments 1: Journal Guidelines: The authors should thoroughly read the instructions provided by the

journal MDPI International Journal of Molecular Sciences and revise the manuscript accordingly. ]

Response 1: We have followed the journal's submission guidelines and made changes and

improvements based on the reviewers' comments. Please see the revised text on a gray background in the manuscript.

Comments 2: [Structure of the Manuscript: Information in the introduction and discussion/conclusion

sections needs to be properly separated. The introduction currently contains too many specific details

about the study results, which should be reserved for the results and discussion sections..]

Response 2: Agree. Based on the reviewers' comments, we have adjusted the details of the experimental

results from the Introduction section to the Results and Discussion section. See page 2, line 91-95

Comments 3: [Selective Toxicity: There is a lack of experiments on healthy cell line to determine the

selectivity of Kuwanon C and justify the relevance of conducting such studies. Including these would

provide a more comprehensive understanding of the compound's safety and potential therapeutic

window.]

Response 3: Based on the reviewers' comments, we chose to evaluate the safety of Kuwanon C at the

healthy cell (HUVEC) level; unfortunately, the results showed that Kuwanon C also inhibited the

proliferation of HUVEC cells (Supplementary Fig. 5), with similar results on tumor cells. In the future,

Kuwanon C could be modified or combined with a nanodrug delivery system to enable Kuwanon C to

target only tumor cells without harming healthy cells. See supplementary material page 5, line 40 and

43

Comments 4: [Figure 1 Adjustments: In Figure 1, the graphs labeled "B" should have the y-axis

labeled as "cell viability in %" instead of "OD." Additionally, IC50 values should be calculated and

presented in the text or a table for all tested cell line. Control compounds (Morin, Paclitaxel, Albanin A)

should also be tested using the MTS assay, and their values reported.]

Response 4: Based on the reviewers' comments, we rearranged the presentation of the effects of

Kuwanon C, Albanin A, Norartocarpetin, Morin, and Paclitaxel on the viability of Hela cells, calculated

the IC50 (Figure 1B), and modified the legend. See page 4, lines 115 and 118

Comments 5: [HeLa Cell Line Justification: The rationale for selecting the HeLa cell line for further

studies is missing. This information should be included to clarify why this particular cell line was

chosen. ]

Response 5: We evaluated the anti-tumor activity of Kuwanon C on cervical cancer cells, breast cancer

cells, and glioma cells, and the results showed that Kuwanon C inhibited the growth of all tumor cells

tested. As we are very concerned about women's reproductive health and cervical cancer is an important

disease that threatens women's reproductive health, we have prioritized the study of the anti-tumor

mechanism of Kuwanon C on Hela cells.

Comments 6: [Image Quality: Figures 2-7 are inferior quality, making it difficult to review the results.

Original images must be provided in a single zip archive as supplementary material files at a sufficiently

high resolution (a minimum of 1000 pixels in width/height, or a resolution of 300 dpi or higher).]

Response 6: Thanks to the reviewers, we have completed the image quality modification of Figures 2-

and 7 to save them at 300 dpi, respectively, and uploaded them separately to the Supplementary

Materials. See page 4, line 115; page 6, line 140; page 8, line 176; page 10, line 214; page 12, line 244;

page 13, line 270; page 15, line 323

Comments 7: [Scale Bars: Figures 2 and 5 are missing scale bars.]

Response 7: We have added scale bars to Figure 2J and Figures 5A and 5B. Also, the mitochondrial

membrane potential fluorescence detection experiments shown in Figures 5A and 5B were done in µg/ml

before. Based on the suggestions of other reviewers, we redone this part of the experiments to make sure

2it matches the full text of the concentration units and to avoid problems with fluorescence overexposure. See page 6, line 140; page 12, lines 243, 246-248

Comments8: [Migration Assay: The migration experiment is flawed because Kuwanon C at 20 µM

inhibited HeLa cell proliferation after 24 hours of incubation. Therefore, the observed effects in the

scratch assay are likely due to inhibited proliferation rather than migration. The experiment should be

redone using a DNA synthesis inhibitor like Mitomycin C, which is the standard practice.]

Response 8: Agree. We revised the cell migration assay based on the reviewer's expert advice,

pretreated Hela cells with 1 μ g/mL mitomycin for an hour, and then carried out the scratch assay.

Figures 2A and 2B display the most recent findings. See page 6, line 140

Comments 9: [Terminology: The term "tumorigenic capacity" is not appropriate for in vitro studies.

The manuscript should use more accurate terminology to describe the findings.]

Response 9: We have modified the tumorigenic capacity to clone formation ability in the

manuscript.See page 6, line 144

Comments 10: [Western Blot Figures: For the membranes shown in Figure 4F, full-length images of

the blots should be provided.]

Response 10: The original Figure 4F WB image has been provided. It should be noted that the target

protein GADD34 and NOXA antibody hybridization in the Figure 4F WB experiment has multiple

non-specific proteins, and the expression is much lower compared to other non-target proteins, which

were previously incubated separately with the PVDF membrane cut, which resulted in no whole figure

being provided before. Therefore, we purchased a new antibody and reran the protein experiment.

However, the GADD34 protein continued to appear in multiple non-specific bands. Consequently, we

split the membrane into two sections for separate exposure, then combined the freshly cut membrane for

the photo. See the original WB image file and Fig 4F. See page 10, line 214

Comments 11: [Microscopy Annotations: Apoptotic/necrotic cells, etc., should be indicated with

arrows in the images (Figure 4F) and described in the figure legend. ]

Response 11: The annotation of the microscope images is complete, and Figure 4A illustrates apoptotic

and necrotic cells with red and white arrows, as illustrated in the legend. See page 10, lines 214, 217

and 218

Comments 12: [Formatting Issues: Section 3.3 is entirely written in italics and should be corrected to

match the formatting of the rest of the manuscript.]

Response 12: Thanks to the reviewers,Section 3.3 Font formatting has been corrected to be consistent

with the full text. See page 18, lines 431-450

3

Reviewer 3 Report

Comments and Suggestions for Authors

The manuscript revealed the role of Kuwanon C in tumor cell viability, cell migration, proliferation, apoptosis, mitochondrial membrane potential, cell cycle and targets through different methods, which are sufficient and important. Interestingly, authors provided some evidences regrading the substrate/target of Kuwanon C which could be further verified. Here are my concerns which can be useful to improve the study.

In figure 1B, the OD values of MTS assay are not consistent at the same dose of kuwanon C using the same cell line. Author should clarify the inconsistency.

In figure 3J, the images only showed the green fluoresecence and lacked the merged images for cell shapes.

In figure 4C, the apoptosis rate is 4.43% which is higher than Albanin A and Norartocarpetin groups. However, the apoptosis staining in fgure 4A showed that almost all cells has red fluorescence, indicating that the apoptosis is more than 90% at 60uM concentration. Does the minor concentration change cause the large discrimination?

In figure 5A and 5B, both levels of red and green fluorescence in control group showed the increase by time extension, indicating the changes in mitochondrial membrane potential. And control is important for comparing different treatment assay, which should have no difference even though the treated timepoints increased from 8h to 24h. Author should provide more explaination.

In figure 7 L-O, authors showed the evidence regarding Kuwanon C binding to CDK1 protein which is a key factor for cell cycle regulation. Also, authors revealed that Kuwanon C inhibits cell proliferation. Does Kuwanon C suppress cell proliferation through CDK1? Authors should clarify the mechanism clearly.

Also, I recommend authors to regulate the result part to being more logical, such as moving the cell cycle section to the last second part.

Comments on the Quality of English Language

Language has minor issues which need to be improved.

Author Response

Comments 1: In figure 1B, the OD values of MTS assay are not consistent at the same dose of

kuwanon C using the same cell line. [The author should clarify the inconsistency. ]

Response 1: Thank you for pointing this out.We agree with this comment. The difference in OD

values between the 1st graph on the left and the next two graphs in Fig. 1B are due to a certain

difference in the number of starting inoculated cells caused by different batches of cell viability

experiments conducted. The reason for the arrangement of experiments in different batches is that we

first carried out experiments on the viability of Kuwanon C on tumor cells and found that Kuwanon

C had the ability to inhibit the viability of tumor cells. Based on the fact that Kuwanon C has two

important isopentenyl groups, we introduced Norartocarpetin, which has differences in the structureof isopentenyl groups, into the subsequent experiments: Moin, Albanin A, and Kuwanon T. To

demonstrate the importance of the isopentenyl moiety of Kuwanon C for exerting anti-tumor cell

viability. Also, based on the comments of other reviewers, we changed the way we presenting the results

(Fig. 1B) and calculated the IC50 values for these substances. (See page 4, lines 115 and 118.)

Comments 2:  In figure 3J, the images only showed the green fluorescence and lacked the merged

images for cell shapes.

Response 2: Agree. Unfortunately, we didn't perform white-light photography during the previous

experiments, which prevented us from taking a photo of the cell shape in Fig. 3J. During this revision,

we redid this part of the experiment and presented the new results in Fig. 3J. (See page 8, line 176.)

page 9, line 189)

Comments 3:  In figure 4C, the apoptosis rate is 4.43%, which is higher than Albanin A and

Norartocarpetin groups. However, the apoptosis staining in Figure 4A showed that almost all cells has

red fluorescence, indicating that the apoptosis is more than 90% at a 60 uM concentration. Does the

minor concentration change cause the large discrimination?

Response 3: According to the results of the mitochondrial membrane potential assay in Figure 5D,

after 60 μ M Kuwanon C treatment of Hela, up to 97.3% of the cells experienced a decrease in

membrane potential. Since the membrane potential change is a hallmark event of early cell apoptosis,

when the Kuwanon C concentration was increased from 50 μM to 60 μM, this small change in the

concentration of Kuwanon C may cause apoptosis to be greatly increased. (See page 10, line 214.)

page 12, line 244)

Comments 4: In figure 5A and 5B, both levels of red and green fluorescence in control group

showed the increase by time extension, indicating the changes in mitochondrial membrane potential.

And control is important for comparing different treatment assay, which should have no difference

even though the treated timepoints increased from 8 hours to 24 hours. author should provide more

explaination.]

Response 4: In the previous membrane potential JC-1 staining assay, overexposure of green

fluorescence in the control group of Fig. 5B and the Morin and Paclitaxel group at 30 μg/ml led to

incorrect results of the experiment, resulting in a large difference between the two control groups.

Together with other reviewers' suggestions that the concentration unit here should be molar,

concentration, we redid the experiment in this revision (removing the Paclitaxel group and

introducing the positive control group in the experiment), and the detailed results are shown in

Figures 5A and 5B. (See page 12, line 244, line 247-248)

Comments 5: In figure 7 L-O, authors showed the evidence regarding Kuwanon C binding to

CDK1 protein, which is a key factor for cell cycle regulation. Also, authors revealed that Kuwanon C

inhibits cell proliferation. Does Kuwanon C suppress cell proliferation through CDK1? Authors

should clarify the mechanism clearly.

Response 5: Many thanks to the reviewers for their questions. According to the experimental results,

obtained so far, it is suggested that Kuwanon C has multiple target molecules in Hela cells, and

CDK1 is one of the target molecules, with a high probability. To get direct evidence that Kuwanon C

inhibits cell proliferation through CDK1, More experiments need to be carried out to prove it, but the

corresponding experiments could not be carried out due to the short time limit of the revision period.

We will continue to explore the clear mechanism in the subsequent studies. The rationale for the

relevant analyses is as follows: The red fluorescent group CY3 was modified for Kuwanon C. Figures

7D7G show that red fluorescent mitochondria were observed, suggesting that Kuwanon C can stably

bind to the mitochondria; as shown in Figure 7N, Kuwanon C-Cy3 has stably bound to mitochondria

and cytoplasmic extracted proteins, in addition to the appearance of bound protein bands at the total

protein . We collected protein bands separately for protein profiling and identified proteins related to

ATP synthesis (e.g., ATP5F1A) in mitochondrial proteins and CDK1 proteins in cytoplasmic

proteins. Molecular docking analyses also showed that Kuwanon C has a very good binding capacity

to CDK1. These results suggest that Kuwanon C is not a single target of action.

Comments 6: Also, I recommend authors regulate the result part to be more logical, such as

moving the cell cycle section to the last second part.

Response 6: Many thanks to the reviewers for the suggestion that the cell cycle section is now

located in the penultimate part of all results. (see pages 12–14, lines 254–280)

Reviewer 4 Report

Comments and Suggestions for Authors

The authors suggest that Kuwanon C disrupts the tumor cell energy metabolism and protein synthesis. 

My concerns:

The molecular mechanism underlying the suggested results of the treatment with Kuwanon C is only partly explained. 

1.       Are only cancer cells impaired by Kuwanon C? 

2.       The effect of Kuwanon C on the cell cycle in Fig. 6A can represent G2/M arrest, as well.

3.       Does Kuwanon C cause mitosis arrest?  The resulting leakage of the mitochondrial membrane due to mitosis arrest causes cell death, Mitotic Catastrophe cell death. 

4.       Some flavonoids inhibit the PARP protein tankyrases (a known target in cancer therapy). Does Kuwanon C inhibit tankyrase activity?  elongation of telomeres in cancer or healthy cells?

5.        The effective concentration range of Kuwanon C is relatively high. What is the effect of  Kuwanon C at this relatively high concentration on healthy tissues and blood cells?

6.       The writing in  Figures   3,5,6,7  is too small. It should be readable.

Author Response

Comments 1: [Are only cancer cells impaired by Kuwanon C? ]

Response 1: Thank you for pointing this out.We agree with this comment. In response to your

other reviewers' shared concern about whether Kuwanon C affects only cancer cells, we conducted

preliminary studies on normal HUVEC cells. Regrettably, the findings indicated that Kuwanon C

also hindered the growth of HUVEC cells, comparable to its effects on tumor cells (Supplementary

Fig. S5). While we did not discover any natural chemicals specifically targeting tumor cells, our

findings demonstrated that at the cellular level of cervical and breast cancer, the anti-tumor impact of

high concentrations of Kuwanon C surpassed that of Paclitaxel at the same concentration. In the

In the future, it is possible to combine Kuwanon C with a nano-drug delivery system to specifically transport Kuwanon C to tumor tissues and cells, resulting in focused anti-tumor therapy and

minimized side effects. [Extensive research is required to achieve this in the future.] The results of the

effect of Kuwanon C on normal cells are placed in the Supplementary Material; see Fig. S5. See

supplementary material, page 5, Lines 40 and 43

Comments 2: The effect of Kuwanon C on the cell cycle in Fig. 6A can represent G2/M arrest, as

well.]

Response 2: Agree. Based on our results (Figs. 6A and 6B), Kuwanon C affects the G2/M phase in

addition to the G1/G0 phase of Hela cells, as suggested by the reviewer comments. (See page 13, line

257

Comments 3: Does Kuwanon C cause mitosis arrest? The resulting leakage of the mitochondria

membrane due to mitosis arrest causes cell death, Mitotic Catastrophe cell death.]

Response 3: Agree. According to the experimental results, Kuwanon C binds to mitochondria and

endoplasmic reticulum and ultimately destroys the normal structure of mitochondria and endoplasmic

reticulum (Fig. 7). Mitochondria and endoplasmic reticulum provide the power of mitosis and the

synthesis of related proteins, which are all closely related to cellular mitosis; therefore, we speculate

that Kuwanon C may affect mitotic arrest, but of course, we still need direct experimental evidence.

We are very sorry! Due to the time constraints of this revision, the relevant experiments could not be

carried out. However, your suggestion is an important direction for us to focus on in the future. See

page 15, line 323

Comments 4: Some flavonoids inhibit the PARP protein tankyrases (a known target in cancer).

therapy). Does Kuwanon C inhibit tankyrase activity? elongation of telomeres in cancer or healthy

cells?]

Response 4: As mentioned by the reviewer, there are related studies with flavonoids showing

inhibition of PARP protease (PMID: 24317580), and we have no experimental evidence on whether

Kuwanon C also inhibits tankyrase activity. Unfortunately, we were unable to carry out this part of

the experiment due to the time constraints of this revision. We will carry out relevant experiments in

subsequent studies.

Comments 5: The effective concentration range of Kuwanon C is relatively high. What is the effect?

of Kuwanon C at this relatively high concentration on healthy tissues and blood cells?

Response 5: As the reviewer mentioned, the effective concentration range of Kuwanon C is relatively

high, and in our previous study, we did not evaluate its safety in vivo or on blood cells. Our existing

studies have found that Kuwanon C's isopentene moiety plays a key role in its antitumor

activity (Fig. 1–Fig. 6), and in future studies, it may be possible to reduce the effective concentration of

the new compounds by increasing the number of isopentene moieties, which may reduce the side

effects on normal tissues or blood cells. We thank the reviewers for their comments, and we will

evaluate the safety of Kuwanon C in vivo in future studies. (See page 4, line 115; page 6, line 140.)

page 8, line 176; page 10, line 214; page 12, line 244; page 13, line 270.

Comments 6: [The writing in Figures 3,5, 6, and 7 is too small.]. [It should be readable.]

Response 6: Many thanks to the reviewers for their suggestions, and in this revision, we provide

high-resolution images to make it readable. High-resolution images of the results were updated in the

manuscript, and a separate zip file of the high-resolution images was uploaded. (See page 8, line 176.)

page 12, line 244; page 13, line 270; page 15, line 323

Round 2

Reviewer 2 Report

Comments and Suggestions for Authors

The revised manuscript has shown significant improvement, and the authors have diligently addressed all of my concerns and suggestions previously raised.